# Associations between physical activity, physical fitness, and body composition in adults living in Germany: A cross-sectional study

Raphael Schilling*, Steffen C. E. Schmidt, Janis Fiedler, Alexander Woll

Institute of Sport and Sport Science, Karlsruhe Institute of Technology, Karlsruhe, Germany

* Raphael.Schilling@kit.edu

## Abstract

### Background & aims

Body composition (BC) changes with age and is associated with morbidity and mortality. A physically active lifestyle influences BC and represents an important predictor of successful aging. To emphasize this, the World Health Organization established activity recommendations for all age groups. We describe BC during adulthood using a cross-sectional sample from a German community and investigate the associations between physical activity (PA), physical fitness (PF), and BC.

### Methods

Data from 329 men and women aged 35 to 86 years were analyzed. PA was measured by questionnaire and classified into sport activity and habitual activity. PF was measured through physical performance tests and BC by bioelectrical impedance analysis. Fat mass index (FMI) and fat-free mass index (FFMI) were calculated to represent height-adjusted BC. Associations between PA, PF, and BC were analyzed using linear regression models.

### Results

For both sexes, strength was positively associated with FFMI ($\male$: ß = 0.313; $\female$: ß = 0.213) and phase angle ($\male$: ß = 0.357; $\female$: ß = 0.409). For FMI, a significant negative association with strength was found only in women (ß = -0.189). Cardiorespiratory fitness showed a negative association with FMI (ß = -0.312) and FFMI (ß = -0.201) for men, while in women a positive association was found for FFMI (ß = 0.186). For coordination, a significant association with FMI was observed only in women (ß = -0.190). Regarding PA only one significant relationship between sport activity and FMI among women (ß = -0.170) was found.

### Conclusions

In our sample, PF was closer related to BC than PA. Strength and cardiorespiratory fitness were the strongest predictors for BC. This supports the World Health Organization's activity

**Funding:** The authors received no specific funding for this work.

**Competing interests:** The authors have declared that no competing interests exist.

recommendations to include both resistance and endurance training in the weekly sports program to maintain a healthy BC.

## Introduction

Aging is characterized by changes in body composition (BC) [1–3]. While fat mass increases, fat-free mass decreases during aging due to the loss of muscle mass, body water, and bone density. Especially among men, the increase in fat mass is more pronounced in the abdominal region, which is associated with a higher risk of mortality [4]. When examining age-related changes in BC, it must be considered that even though the body-mass-index (BMI) may remain constant, an increase in fat mass with age is common [5]. The results of a recent review reveal, that an increased percentage of fat, an increased hip circumference, or an increased waist-to-hip or height ratio increases the risk for morbidity in normal-weight individuals [6]. This indicates that both body weight and BMI are not always suitable parameters to assess potential negative anthropometric changes with age [5, 7]. Therefore, appropriate methods such as bioelectrical impedance analysis have been used to assess BC more specifically [8]. In addition to fat mass and fat-free mass, bioelectrical impedance analysis also measures phase angle (PhA). This parameter provides information about the capacitive resistance of the body which is directly linked to cell mass and cell integrity and thus represents an important indicator in the clinical setting [9]. A review's findings imply that PhA serves as a reliable predictor of mortality risk [10]. PhA decreases with age [11] and is a prognostic parameter for various diseases [12]. The results of a recent review even suggest that PhA could be used as a predictor for the development of age-related sarcopenia [13]. Sarcopenia represents a major health problem for the aging population and is defined by a decrease in muscle mass and muscle function [14]. The causes of sarcopenia are various and can be linked to an inactive lifestyle [15], malnutrition [16], chronic diseases [17], and hormonal changes [18], among other factors. The effects of sarcopenia can be severe, increasing the risk of falls and fractures [19], limiting mobility, and affecting the quality of life [20].

The changes in BC with age are primarily due to an imbalance between energy intake and energy expenditure [21]. Thus, physical activity (PA) plays a crucial role in the process [22]. Many studies show that PA is associated with lower obesity rates and weight loss in adults aged 18 up to 85 years [23]. Accordingly, PA has a significant impact on health [24] and is a protective factor against the development of many non-communicable diseases [25]. Therefore, the World Health Organization (WHO) has established PA recommendations for different age groups, explicitly including both, endurance and strength training for individuals in middle and later adulthood [26]. Closely related to PA [27], physical fitness (PF) also plays a major role in healthy aging [28]. While PA is defined by the movement of skeletal muscles which leads to energy expenditure, PF is expressed in daily life through various skills and routines [27]. The results of a longitudinal study even suggest that PF dominates PA in its influence on health [29]. In addition, muscle strength seems to have a decisive influence on well-being [30]. Accordingly, an age-related decline in muscle strength is associated with an increased risk of falls [31] and can lead to functional dependence and disability [32].

Nowadays, external influences like digital and social media as well as new technologies change our daily routines, PA and lifestyle, making it important to analyze which behaviors are beneficial for successful aging. In addition, the world's older population is growing dramatically [33]. With an aging population and the associated changes in BC, it is important to

determine how PA, PF, and BC are associated with aging and themselves. Current cross-sectional studies already indicate that both PA [34–38] and PF [39, 40] are associated with BC in adulthood. Most of these studies measure BC using bioelectrical impedance analysis [37, 38] or dual-energy X-ray absorptiometry [34, 35, 39, 40]. However, only a few studies use the fat mass index (FMI, as of fat mass (kg)/height2 (m)), and fat-free mass index (FFMI, as of fat-free mass (kg)/height2 (m)) to describe BC. Recent literature suggests that there is a rationale for using FFMI and FMI instead of body fat percentage for tracking healthy BC, especially when comparing data gathered through different technical procedures or identifying clinical phenotypes like Sarcopenia. Changes in BC are not necessarily clearly evident via the percentage of body fat, as both an increase in fat mass and a decrease in fat-free mass can result in the same percentage of body fat. For example, people with the same percentage of fat mass who differ in height may have a different nutritional state and/or fitness. In contrast, FMI and FFMI consider the amount of fat mass and fat-free mass in relation to body size [41]. Additionally, only a few studies look at the interdependency of all three constructs: PA, PF, and BC. In this context, the WHO states specifically that more research is needed on the relationship between PA and health outcomes [26].

In our study, we used a cross-sectional design to describe the age-related development of BC among adults and to examine the sex-specific associations between PA, PF, and BC.

## Materials and methods

### Design

All data originates from a community-based, longitudinal German study [42] with currently six measurements in 1992, 1997, 2002, 2010, 2015, and 2021. Our cross-sectional analysis refers to the 2021 measurement point.

### Sample

For the 2021 measurement, n = 666 participants that participated in earlier waves were invited to join the study again and n = 300 did so (response: 45,1%). In addition, a new cross-sectional cohort of 300 35-year-olds (response: 14,7%) and 300 55-year-olds (response: 23,3%) were randomly selected from the residents' registration offices in Bad Schönborn, Germany. In total, 430 adults participated in 2021 and the age range was 35 to 86 years. Participation was voluntary and participants provided their written consent to participate in the study. The applied protocols were approved by a scientific advisory council, the Schettler Clinic, Bad Schönborn, Germany as well as the ethics committee of the Karlsruhe Institute of Technology. We strictly followed ethical guidelines from the German Psychological Society. The 2021 measurement took place during a low-incidence time frame of the COVID-19 pandemic in Germany from June 21st, 2021 to July 29th, 2021. For our analyses, 88 subjects were excluded due to missing data in the variables of interest, and 13 subjects were excluded due to extreme values in the study variables. Finally, a sample of 329 participants was examined in our study.

### Measures

In the following, we describe the methods that were used to obtain PA, PF, and BC data. More details on the study procedures and additional measures have been published previously [43].

### Physical activity

PA was assessed via digital questionnaires that were filled in on computers in the survey center. The questionnaire was proofed in relation to reliability (test-retest after two weeks: r>.90 and

Cronbach's α = .94), factorial validity, and measurement invariance [44]. The questionnaire differentiates between sport activity and habitual activity. Sport activity includes all physical-sporting activities that can be assigned to sports or exercising. In addition to the question "Do you engage in sports or gymnastic exercises?", the types of sports were asked, as well as the minutes per week and the number of weeks per year in which the activity is pursued. The maximum weekly duration was set to 900 minutes per sport and values above 900 minutes for one sport activity were trimmed to 900. The total minutes of sport activity is the sum of all minutes in the specified sports, multiplied by the concordant weeks per year, divided by 52. For sports that do not cause a noticeable increase in energy expenditure by large skeletal muscle (e.g. chess, esports), the duration was set to zero. To estimate habitual activity, the minutes of weekly walking and cycling for transportation as well as other exhausting habitual activities such as gardening were collected.

Daily walking was determined by asking about the distance walked on a typical weekday. Possible answers were: "I almost never walk 0km", "Less than 1km/day (only in the house) 0.5km", "1–2 km/day (in the house and shorter walking distances) 1.5km", "3–5 km/day (longer walking distances away from home) 4km", "6–9 km/day 7km", "10km/day and more 12km". To obtain an average duration in minutes, the resulting average distances were multiplied by a factor of 5.25*60. The factor 5.25 results from the average speed of walking for pleasure according to Ainsworth and colleagues [45]. Daily cycling was raised through a question of whether the participant uses a bicycle for transportation and if the answer was yes, a second item asked about how many minutes they usually cycle daily. Experience showed that for most participants, the minutes of daily cycling can be remembered or added up much easier than the duration of daily walking. Exhausting leisure activities were asked in the form of two items, "Do you perform other physically demanding leisure activities (e.g., gardening)?" and "Overall, how much time do you spend doing this (other than exercising) in minutes per week?". Here, we did not ask about the daily minutes, but the weekly minutes, since these activities often take place far more irregularly compared to the distances traveled. Finally, converting strenuous leisure activities into minutes per day and adding the minutes spent walking and cycling resulted in an index of habitual activity for all participants.

## Physical fitness

Twelve motor performance tests were used to assess PF in 2021. All tests were performed during a single session chronologically after the assessment of BC and were supervised, following a standardized test protocol with acceptable reliability [46]. Cardiorespiratory fitness (CRF) was measured by maximal oxygen uptake ($VO_2$max) estimated from time for completion and heart rate during a 2-km walking test [47]. Strength was assessed by a handgrip (digital hand dynamometer, 198 lbs, GRIPX, China) and a jump-and-reach test. Coordination was measured by standing on one leg with closed eyes, whilst moving the second leg in circles, and three test items with balls: Throw against a wall and catch (exteroceptive, ballistic), throw with rotation and catch (exteroceptive, ballistic, pressure of time), as well as a task where you hold a ball between your legs with one hand in front of the thigh and the other hand behind the other thigh, then releasing and catching the ball with changing the grip five times (interoceptive, tactile-ballistic) [46]. Every task was judged according to standardized rules by the instructors, differentiating among "2—task solved easily", "1 = task solved with problems" and "0 = task not solved". Finally, a coordination index (0–8) of the added-up points was calculated for each participant. Flexibility was measured by a sit-and-reach test and a trunk side bending test, as well as an index for muscle shortening derived from six three-scaled (2 points = no restrictions, 1 point = slight restrictions, 0 points = major restrictions) items for shoulder neck mobility (left/right), hamstring (left/right), and rectus femoris (left/right) muscle extensibility. The index for

muscle shortening ranged from 0 to 12 and was obtained by adding up the achieved points in shoulder neck mobility (left/right), hamstring (left/right), and rectus femoris (left/right).

For handgrip, jump-and-reach, sit-and-reach test, and coordination items, the best of two trials was used in the analyses, all other items were performed one time. The test battery was developed in cooperation with the UKK Institute in Tampere, Finland [46]. Because items with different units were analyzed, a Z-score transformation was used for VO$_2$max, handgrip-strength, jump-and-reach-heights, achieved points in the coordination test, and achieved points in the muscle shortening test battery. These items were standardized according to Woll et al. [43] based on male participants aged between 33 and 36 years in 1992. Thus, the performance of 35-year-old males in 1992 is Z = 100 in all test items of motor performance. The Z-value transformation results as follows:

$$Z = 100 + \text{raw value} - \frac{\bar{\text{x}}}{\bar{\text{s}} \bullet} 10$$

$\bar{\text{x}}$ = mean value of the 33-36-year-old men at the first time of measurement in 1992
$\bar{\text{s}}$ = standard deviation of the 33-36-year-old men at the first time of measurement in 1992

Finally, indices for strength (mean Z-value from handgrip and jump-and-reach), CRF (Z-value of the VO$_2$max), coordination (Z-value of the coordination index), and flexibility (Z-value of the flexibility index) were calculated and used for the latter analyses.

## Body composition

We measured body height with a fixed stadiometer (seca 213, seca gmbh, Germany) and waist circumference by measuring tape through trained personnel. The seca medical body composition analyzer (seca mBCA 515) was used to measure body weight and BC according to the international ESPEN standards [48]. According to ESPEN standards, the absence of drugs but not fasting was a prerequisite and participants were invited to join the study with normal nutritional status. From bioelectrical impedance analysis, we used PhA and calculated FMI and FFMI. FMI was calculated as fat mass (kg)/height$^2$ (m) and FFMI as fat-free mass (kg)/height$^2$ (m). Fat mass and fat-free mass were derived from the seca formulas with age- and sex-related in-house reference data for resistance and reactance, which are not publicly accessible. The formulas and approach to estimate total body water and body fat were validated against a four-compartment model and a two-compartment model (air-displacement plethysmography, dual-energy X-ray absorptiometry, and deuterium dilution) [49]. The validity of estimating segmental skeletal muscle mass was tested separately against magnetic resonance imaging and dual-energy X-ray absorptiometry and showed that the magnetic resonance imaging-based measurement results of the seca mBCA are more accurate than those of a dual-energy X-ray absorptiometry measurement [50].

## Statistical analysis

The IBM© SPSS© Statistics package (Version 28.0) was used for the analysis. To present basic descriptive statistics, mean values and standard deviations were calculated for all study variables. As previously described, subjects with extreme values were excluded. In our work, extreme values for men and women, respectively, were defined as data points that were more than 3 standard deviations away from the upper or lower limit of the interquartile range. Pearson product-moment correlation coefficients (r) were performed to examine the relationship between PA, PF, BC, and age. To visualize the relationships between FMI, FFMI, and PhA with age, scatter plots were created for men and women. Regression models were performed to examine the cross-sectional associations of age, PA, and PF with FMI, FFMI, and PhA. The

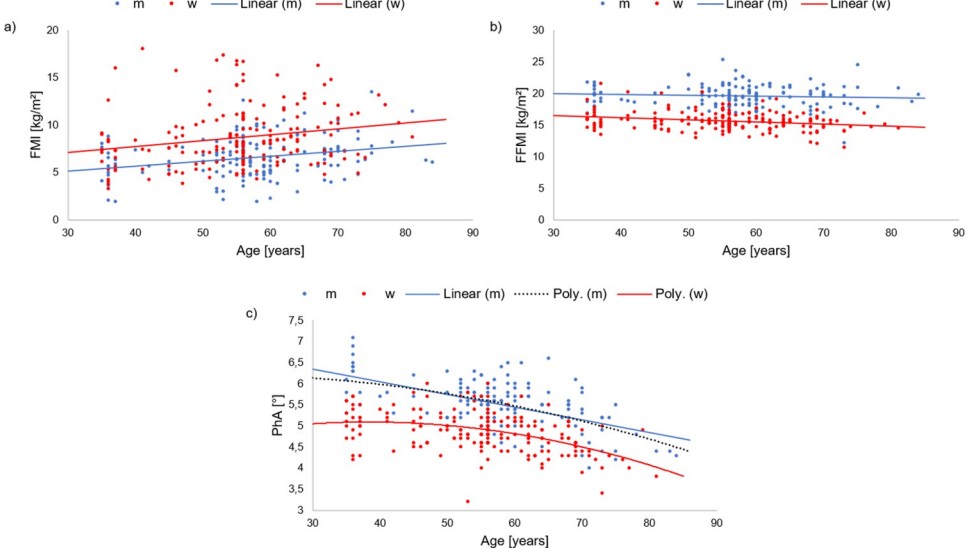

**Fig 1.** Scatter plots showing the relationship between age in years and a) FMI, b) FFMI as well as c) PhA.

final models consisted of age (Model 0), sport activity, habitual activity (Model 1), strength, coordination, flexibility, and CRF (Model 2) with FMI, FFMI, and PhA as the dependent variables. Age was also included as an additional predictor in Model 1 (Model 1.2) and Model 2 (Model 2.1). Lastly, in Model 3, all predictors were considered in one model. The analyzes were carried out separately for each sex. Since we do not assume a linear change of BC during aging, polynomial regression models were calculated for all parameters of BC and age. From those analyses, only PhA among women showed a significant negative quadratic decline. We show this polynomial curve progression in Fig 1C and added the quadratic age coefficient in Table 3. Age was grand mean centered at the average age of all participants. In the end, six models each were calculated for men and women for the dependent variables FMI, FFMI, and PhA. Statistical significance was set a priori at the level of $p \leq 0.05$.

## Results

### Descriptive statistics

A total of 329 adults (183 women, 146 men) participated in this study. The main characteristics of the sample are presented in Table 1. Despite the higher body weight in men, women had a higher FM. FFMI and PhA were in a higher range for men while FMI was higher in women. Men showed higher values for habitual activity, while women engaged more often in sports activities. Concerning PF, there were higher values in strength, coordination, and CRF in men, while women performed better in flexibility.

### Relations with age

Fig 1 shows the associations between age and a) FMI, b) FFMI, and c) PhA. The associations between age and PF are presented in the S1 Fig. For both sexes, the scatter plots demonstrate that persons with higher age tended to have higher FMI values, lower FFMI values and lower PhA values. While FMI increased almost equally in both sexes, FFMI showed a greater decrease in women. An age-related decline for PhA was observed in men and women, whereby among women, the decline showed a significant quadratic proportion.

**Table 1. Descriptive characteristics of the participants divided by sex.**

| | Men (n = 146) Mean (SD) | Women (n = 183) Mean (SD) |
|---|---|---|
| *Anthropometrics and body composition* | | |
| Age (years) | 57.7 (10.5) | 55.3 (10.6) |
| Height (cm) | 178.0 (7.1) | 164.5 (5.9) |
| Weight (kg) | 82.8 (10.9) | 65.8 (11.6) |
| BMI (kg/m$^2$) | 26.2 (3.3) | 24.3 (4.1) |
| FM (kg) | 20.8 (6.4) | 23.4 (8.5) |
| FFM (kg) | 62.0 (6.9) | 42.4 (4.9) |
| FMI (kg/m$^2$) | 6.6 (2.1) | 8.7 (3.2) |
| FFMI (kg/m$^2$) | 19.6 (1.8) | 15.7 (1.5) |
| PhA (°) | 5.5 (0.6) | 4.9 (0.5) |
| *Physical activity* | | |
| HA (min/week) | 458.1 (301.7) | 431.8 (281.7) |
| SA (min/week) | 171.2 (159.7) | 191.3 (167.0) |
| *Physical fitness* | | |
| Strength (z-Score[a]) | 84.9 (11.7) | 62.6 (9.2) |
| Coordination (z-Score) | 90.3 (11.0) | 85.2 (11.4) |
| Flexibility (z-Score) | 99.0 (9.9) | 106.3 (8.6) |
| CRF (z-Score) | 95.3 (13.3) | 84.6 (4.7) |

Abbreviations: n, sample size; SD, standard deviation; cm, centimeter; kg, kilogram; BMI, body-mass-index; m, meter; FM, fat mass; FFM, fat-free mass; FMI, fat mass index; FFMI, fat-free mass index; PhA, phase angle; °, degree; HA, habitual activity; min, minutes; SA, sport activity; CRF, cardiorespiratory fitness.
[a]z-standardized variable.

## Associations between physical activity, physical fitness, and body composition

Regression models for FMI, FFMI, and PhA are presented in Tables 2 and 3. The models were calculated stratified by sex. An overview of the different correlations among the presented variables can be found in the S1 Table.

Among men, only age [Model 0, age: ß = 0.260; p = .002] and CRF [Model 2, CRF: ß = -0.312; p = .002] showed significant associations with FMI. In Model 2.1 the significance of the effect of age [Model 2.1, age: ß = 0.031; p = .776] disappeared when PF was added. Habitual activity [Model 1, HA: ß = -0.014; p = .872] and sport activity [Model 1, SA: ß = -0.008; p = .932] did not influence FMI among men in our sample. FFMI was significantly associated with PF [Model 2: R$^2$ = .084], with strength [Model 2, strength: ß = 0.313; p = .002] being the strongest predictor. Besides strength, CRF also exerted an influence on FFMI [Model 2, CRF: ß = -0.201; p = .044], which increased in Model 3 [Model 3, CRF: ß = -0.261; p = .020]. PhA decreased significantly with age among men [Model 0, age: ß = -0.558; p = < .001]. Among PF variables, PhA was predicted primarily by strength [Model 2, strength: ß = 0.357; p = < .001]. In model 2.1, that is when age was added to model 2, a decrease in the regression coefficients of age [Model 0, age: ß = -0.558; Model 2.1, age: ß = -0.411] and strength [Model 2, strength: ß = 0.357; Model 2.1, strength: ß = 0.198] could be observed.

Among women, age [Model 0, age: ß = 0.209; p = .004], sport activity [Model 1, SA: ß = -0.170; p = .024], strength [Model 2, strength: ß = -0.189; p = .040], and coordination [Model 2, coordination: ß = -0.190; p = .019] showed a significant relationship to FMI. The effect of age [Model 2.1, age: ß = -0.029; p = .753] disappeared in Model 2.1. The activity model [Model 1: R$^2$ = 0.027] explained less variance of FMI than the fitness model [Model 2: R$^2$ = 0.112].

**Table 2. The adjusted total coefficient of determination (R² adj.) and the unstandardized regression coefficients (β) of age, PA, and PF in relation to FMI, FFMI, and PhA with men.**

| Men | FMI [kg/m²] | | FFMI [kg/m²] | | PhA [˚] | |
|---|---|---|---|---|---|---|
| | R² (adj.) | ß | R² (adj.) | ß | R² (adj.) | ß |
| *Model 0* | 0.061 | | -0.001 | | 0.307 | |
| Age (year) | | 0.260* | | -0.076 | | -0.558* |
| *Model 1* | -0.014 | | -0.006 | | 0.014 | |
| HA (min/week) | | -0.014 | | -0.008 | | 0.152 |
| SA (min/week) | | -0.008 | | 0.091 | | 0.037 |
| *Model 1.1* | 0.050 | | -0.005 | | 0.338 | |
| Age (year) | | 0.267* | | -0.092 | | -0.576* |
| HA (min/week) | | 0.003 | | -0.013 | | 0.115 |
| SA (min/week) | | -0.052 | | 0.106 | | 0.133 |
| *Model 2* | 0.117 | | 0.084 | | 0.237 | |
| Strength (z-Score[a]) | | -0.066 | | 0.313* | | 0.357* |
| Coordination (z-Score) | | -0.036 | | -0.143 | | 0.025 |
| Flexibility (z-Score) | | -0.009 | | 0.167 | | 0.058 |
| CRF (z-Score) | | -0.312* | | -0.201* | | 0.164 |
| *Model 2.1* | 0.111 | | 0.078 | | 0.324 | |
| Age (year) | | 0.031 | | -0.028 | | -0.411* |
| Strength (z-Score) | | -0.055 | | 0.302* | | 0.198* |
| Coordination (z-Score) | | -0.034 | | -0.145 | | 0.004 |
| Flexibility (z-Score) | | -0.010 | | 0.168 | | 0.061 |
| CRF (z-Score) | | -0.301* | | -0.211* | | 0.012 |
| *Model 3* | 0.142 | | 0.085 | | 0.354 | |
| Age (year) | | 0.028 | | -0.072 | | -0.453* |
| HA (min/week) | | -0.016 | | -0.046 | | 0.105 |
| SA (min/week) | | 0.010 | | 0.152 | | 0.137 |
| Strength (z-Score) | | -0.054 | | 0.300* | | 0.184* |
| Coordination (z-Score) | | -0.033 | | -0.127 | | 0.026 |
| Flexibility (z-Score) | | -0.009 | | 0.176 | | 0.067 |
| CRF (z-Score) | | -0.304* | | -0.261* | | -0.031 |

Abbreviations: FMI, fat mass index; kg, kilogram; m, meter; FFMI, fat-free mass index; PhA, phase angle; ˚, degree; R² (adj.), adjusted coefficient of determination; ß, beta coefficient; HA, habitual activity; min, minutes; SA, sport activity; CRF, cardiorespiratory fitness.

[a] z-standardized variable.

* ß is significant at the 0.05 level.

While age did not contribute to the prediction of FFMI in men, it did among women [Model 0, age: ß = -0.238; p = .001]. Fitness in the form of strength [Model 2, strength: ß = 0.213; p = .025] and CRF [Model 2, CRF: ß = 0.186; p = .038] also showed a significant impact on FFMI. By adding age into the fitness model of the females, a decrease in the regression coefficients of age [Model 0, age: ß = -0.238; Model 2.1, age: ß = -0.190], strength [Model 2, strength: ß = 0.213; Model 2.1, strength: ß = 0.150], and CRF [Model 2, CRF: ß = 0.186; Model 2.1, CRF: β = 0.143] could be observed. Fig 1 shows that the relationship between PhA and age was more parabolic for women compared to men. Model 0 shows that the regression coefficients of both, age [Model 0, age: ß = -0.476; p = < .001] and age² [Model 0, age²: ß = -0.182; p = .010] contributed significantly to the prediction of PhA. In addition to age, strength was also an important predictor of PhA in women [Model 2, strength: ß = 0.409; p = .002]. PA, on the other hand, did not contribute to the prediction [Model 1: R² = -0.004]. By adding age into the

**Table 3. The adjusted total coefficient of determination ($R^2$ adj.) and the unstandardized regression coefficients (β) of age, PA, and PF in relation to FMI, FFMI, and PhA with women.**

| Women | FMI [kg/m²] | | FFMI [kg/m²] | | PhA [°] | |
|---|---|---|---|---|---|---|
| | $R^2$ (adj.) | ß | $R^2$ (adj.) | ß | $R^2$ (adj.) | ß |
| *Model 0* | 0.039 | | 0.051 | | 0.196 | |
| Age (year) | | 0.209* | | -0.238* | | -0.476*c; -0.182*c2 |
| *Model 1* | 0.027 | | -0.007 | | -0.004 | |
| HA (min/week) | | -0.066 | | -0.006 | | 0.078 |
| SA (min/week) | | -0.170* | | -0.063 | | 0.018 |
| *Model 1.1* | 0.073 | | 0.043 | | 0.197 | |
| Age (year) | | 0.226* | | -0.234* | | -0.482*c; -0.179*c2 |
| HA (min/week) | | -0.067 | | -0.005 | | 0.071 |
| SA (min/week) | | -0.188* | | -0.044 | | 0.056 |
| *Model 2* | 0.112 | | 0.061 | | 0.248 | |
| Strength (z-Score) | | -0.189* | | 0.213* | | 0.409* |
| Coordination (z-Score) | | -0.190* | | -0.114 | | 0.070 |
| Flexibility (z-Score) | | -0.123 | | -0.084 | | -0.036 |
| CRF (z-Score) | | 0.049 | | 0.186* | | 0.134 |
| *Model 2.1* | 0.116 | | 0.077 | | 0.267 | |
| Age (year) | | -0.029 | | -0.190* | | -0.220*c; -0.117c2 |
| Strength (z-Score) | | -0.199* | | 0.150 | | 0.340* |
| Coordination (z-Score) | | -0.197* | | -0.158 | | 0.037 |
| Flexibility (z-Score) | | -0.124 | | -0.093 | | -0.038 |
| CRF (z-Score) | | 0.043 | | 0.143 | | 0.068 |
| *Model 3* | 0.149 | | 0.069 | | 0.265 | |
| Age (year) | | 0.006 | | -0.180 | | -0.235*c; -0.119c2 |
| HA (min/week) | | -0.055 | | -0.026 | | 0.045 |
| SA (min/week) | | -0.190* | | -0.046 | | 0.057 |
| Strength (z-Score) | | -0.215* | | 0.147 | | 0.343* |
| Coordination (z-Score) | | -0.185* | | -0.155 | | 0.032 |
| Flexibility (z-Score) | | -0.127 | | -0.094 | | -0.036 |
| CRF (z-Score) | | 0.087 | | 0.156 | | 0.049 |

Abbreviations: FMI, fat mass index; kg, kilogram; m, meter; FFMI, fat-free mass index; PhA, phase angle; °, degree; $R^2$ (adj.), adjusted coefficient of determination; ß, beta coefficient; HA, habitual activity; min, minutes; SA, sport activity; CRF, cardiorespiratory fitness.

[a] z-standardized variable.

* ß is significant at the 0.05 level.

[c] age-centered.

[c2] age-centered and squared (added as a second variable in addition to age).

fitness model of the females, a decrease in the regression coefficients of age [Model 0, age: ß = -0.476, age²: ß = -0.182; Model 2.1, age: ß = -0.220, age²: ß = -0.117] and strength [Model 2, strength: ß = 0.409; Model 2.1, strength: ß = 0.340] could be observed similar to the males.

## Discussion

The goal of our study was to describe BC during adulthood using a cross-sectional sample from a German community and to investigate the associations between PA, PF, and BC under consideration of age. Our results confirm a relationship between age and FMI, FFMI, as well as PhA. FMI increased with age for both sexes, whereas PhA decreased. FFMI decreased

significantly among women, but not among men and the decrease of PhA among women showed a significant quadratic term, indicating an even increasing decline with increasing age. Furthermore, the results show that there was an association between different parameters of PF and BC for both sexes. In particular, strength and CRF were associated with BC. Regarding PA, only one significant relationship between sport activity and FMI was found among women. No associations with BC could be detected for habitual activity.

## Body composition during adulthood

In our study, the highest associations for both sexes were found for age and PhA. This is supported by the results of a study with a comparable sample that determined that age is a strong predictor for PhA [51]. PhA is a health parameter of growing interest [52]. It is used in the clinical setting [53] and decreases with a variety of diseases [54]. In a recent review, it was shown that the development of sarcopenia can be associated with an age-related decrease in PhA [13]. Accordingly, studies showed that PhA can be used to detect sarcopenia [55]. To counteract age-related declines in PhA, it is essential to understand the mechanisms behind the change of PhA during the lifespan [54]. Regarding the relation between age and FFMI, only women showed a significant association in our study. In addition, women showed a significant, negative quadratic interrelation between PhA and age. Muscle strength is directly related to PhA [13, 56–58] and studies showed that it decreases more rapidly with increasing age in both sexes [59–61]. An explanation for why we found a quadratic relationship only for women might be an insufficient sample size and age-range as well as relatively large intra-individual differences, especially in rather active samples like the present with on average more than 3 hours of sport activity per week. Studies also found that the rate of age-related loss of strength is higher than the decrease in muscle mass [32], which could explain why in our study, higher associations were found between age and strength compared to age and FFMI. Low muscle strength can lead to functional impairment and a higher risk of mortality [62]. Studies also claim that the negative health effects due to a decrease in strength are larger than those due to a decrease in muscle mass [32, 63]. A decline in muscle strength called "Dynapenia" should be considered independently of sarcopenia for the reasons given above [64]. Our results suggest that PhA can be used as an indicator to detect Dynapenia more reliably than FFMI when direct strength tests are not applicable, for example in a clinical setting, among the oldest age groups, or when a series of measurements on a daily basis is planned. We also derive from our results that strength training in adulthood is of high importance and should be used to increase muscle mass and more importantly, muscle strength.

Although the calendar age variable often shows strong correlations with parameters of lifestyle, fitness, and health, it should be noted that, from a theoretical point of view, time cannot be used to causally explain developmental trajectories [65]. There are always biological, psychological, or exterior processes and events that are responsible for age-related changes [66]. For example, a review concluded that a decrease in resting energy consumption caused by a decrease in organ mass and organ metabolism leads to increasing changes in BC with increasing age [21]. Thus, the effects of age should decrease when more explaining variables that influence the outcomes are added. This is also evident in our results. For both sexes, the regression coefficients of age decreased by adding fitness parameters to the model (Model 2.1).

## Associations between physical activity and body composition

In our cross-sectional analyses, there is no evidence that the total volume of sport activity or habitual activity is directly linked to BC. Higher self-reported activity levels do not directly entail a healthy BC in the form of low-fat mass and high levels of fat-free mass. This

assumption is supported by the results of a longitudinal analysis of the present study's measurement points before 2021 [29]. It revealed that an increase in the activity level does not lead to health benefits, such as a reduction in BMI. Over 18 years, BMI increased by 1.2 points per year and PF decreased by 0.8 points per year, although the average PA levels increased slightly by 0.13 weekly metabolic equivalent (MET) hours per year [29]. In a study that measured PA through both, self-report and accelerometry, muscle strength was mainly explained by sex (40–74% variance explained), age (6–44%), and BMI (2–16%) with only 1 to 3 percent of variance explained by PA [67].

It is assumed that PF acts as a mediator between PA and different health parameters such as overweight and subjective as well as diagnosed orthopedic and psychosomatic impairments [29, 68]. That means PA that does not lead to significant changes in PF may not result in health benefits and that amounts and intensities of PA that lead to a decrease in PF (overtraining) can also lead to a decline in health. There is a broad scientific agreement that intensity, as well as type, execution, and periodization of PA, is decisive for its effects on the human body [69, 70]. While higher exercise intensities might be associated with an increase in PF [71], low-intensity physical activities do not seem to be correlated with most parameters of PF [72]. Studies also showed that the dose-response is higher for the total volume of PA and PF compared to the total volume of PA and health measures like all-cause mortality, stroke, and several coronary heart disease risk factors [73]. For BC and PA, similar dose-response principles are assumed. Especially moderate-to-vigorous physical activities are related to the parameters of the BC [35, 74]. However, smaller effects of PA on BC could also be found for light PA [75]. In our study, we found an association between sport activity and FMI only among women. A larger emphasis given on health and body weight [76, 77] instead of physical improvement, muscle mass, and strength during the choice and execution of sports as well as the linked lifestyle and eating patterns among women [78] could be one reason for this outcome. Reasons for a missing association between PA and FMI among men may be the underlying motive of exercising. A potential loss of fat mass may be prevented by an increase in appetite and food intake when exercising primarily for improving strength and muscle mass which is usually preferred by men [79].

A recent review found that in females, the total volume of PA plays a more important role in health benefits than intensity [80]. Especially concerning metabolic syndrome, which is also defined by an immoderate amount of body fat [81], two studies identified a positive association with the volume of PA [82, 83] whereas another study found positive effects only between moderate to vigorous PA and metabolic syndrome [84]. According to this research, women should choose a higher volume of PA to achieve health effects such as a decrease in FM. These circumstances might be another reason why we found a significant negative relationship between the volume of sport activity and FMI in women. In addition to the combination of load intensity and volume of PA, the duration of every single bout of PA also seems to have a decisive influence. In a recent review, it was concluded that despite high intensities and an associated increase in physical performance, short exercise bouts between 60 and 240 seconds did not lead to a reduction in body fat [85].

In conclusion, our results regarding PA and BC indicate that in women, a higher volume of sport activity is related to a lower FMI value. Discussing the results in the context of the current state of research, we assume that high amounts of PA are more effective in women, especially in reducing FM. Aerobic activities are effective for the reduction of body fat but have a lower impact on fat-free mass or PhA. Strength training could be decisive here. Improving BC through PA, therefore, requires a combination of endurance and weight training, as well as a periodization of training. This supports the activity recommendations of the WHO.

## Associations between physical fitness and body composition

Strength and CRF, as parameters of physical performance, were favorably associated with FMI, FFMI, and PhA in both sexes.

Strength showed similar results for both, men and women. Accordingly, higher strength scores were associated with higher scores for FFMI and PhA. This supports the results of a recent study of older people in New Zealand. Here, a positive association between muscle strength and the appendicular skeletal muscle mass index was found for both sexes [39]. In addition to muscle mass, muscle strength also correlated with PhA in our study, which has also been found in several studies [13, 56–58]. It should be considered that PhA is influenced by body cell mass in addition to factors such as sex and age [11, 86]. Muscle mass is an essential component of body cell mass [9]. Therefore, men show on average a higher PhA compared to women due to their higher muscle mass [2, 86, 87] and there is a decrease in PhA with an age-related decrease in muscle mass [87]. The fact that PhA is related to muscle mass has already been supported by several studies [51, 57, 58, 88]. Strength training in particular seems to have a positive effect on PhA [89]. However, resistance training should be performed not only to increase muscle strength but also to counteract age-related declines that can lead to sarcopenia and frailty. Both conditions are associated with a higher risk of mortality [90]. A recent review has revealed that resistance training provides a significant impact on the progression and outcome of sarcopenia and frailty. Both, muscle strength and muscle mass increase with resistance training, even in people who are already suffering from the disease [91]. Integrating our results into the state of research, we can conclude that there is a close association between muscle strength, muscle mass, and PhA for both sexes and that resistance training is probably an essential form of training to positively influence these.

Although it is known that isolated endurance training to improve CRF has positive effects on the musculoskeletal system [92], the effects on BC are lower compared to resistance training. In a study among elderly persons in Spain, a cross-sectional design demonstrated a positive correlation coefficient between $VO_2max$ and body weight in women [93]. According to the authors, an increase in $VO_2max$ was combined with an increase in muscle mass, which explains the gain in body weight. Considering our results and the state of research, we conclude that the effect of exercising on weight, BMI, and BC shows considerable intra-individual differences and therefore, ultimately depends on the characteristics of the sample. Whereas normal-weight individuals may gain weight, as well as muscle mass and fat mass when exercising, individuals with overweight may lose weight, fat, and even muscle mass. For example, in our study, a negative association was shown between CRF and FMI for men, but not for women, which is confirmed by other studies [93, 94]. Also, in younger adult males, it was found that regardless of whether higher muscle mass or fat mass was present, $VO_2max$ was negatively associated with body weight [94]. Possibly the hormonal differences between the sexes [80] lead to different adaptations through sports activities. Certain sports activities could also induce a higher exertion in women than in men [93] which could rather lead to positive effects such as a faster increase in muscle mass. As mentioned before, body fat responds predominantly to aerobic exercise while resistance training exerts a larger effect on muscle mass [35], which could explain why there was no relationship between strength and FMI in men, while a negative relationship between CRF and FMI has been recorded. In women, on the other hand, there was a negative association between strength and FMI. This is consistent with the results of a study on middle-aged and older adults [40].

Although obesity is related to a higher absolute muscle torque and power in selected muscles like the quadriceps, obese individuals show lower relative (normalized to body mass) torque and power as well as lower fatigue resistance in isokinetic voluntary fatigue protocols [95].

This may be due to a decrease in muscle fiber recruitment caused by excessive fat mass [40]. Studies also showed that a decrease in muscle strength is associated with a reduction in coordination [96]. This may explain why we found a significant relationship between coordination and FMI in women but not in men, as women show higher values for FMI and simultaneously, lower strength and muscle mass to potentially compensate for overweight in gross-motor coordination tasks. However, a decrease in the activity level also has a negative effect on coordination ability [97].

In summary, the findings reveal that especially strength and CRF are related to BC. This may imply that both strength and CRF capacity should be increased through PA to maintain positive effects on BC and other dimensions of PF and to counteract age-related declines.

## Strengths and limitations

There are limitations to our study. First, due to the chosen cross-sectional design no causal statements can be made. The results thus provide information at a single point in time. Second, nutrition was not recorded. Nutrition is considered one of the main influencing factors for BC [98] and may influence the relationship between PA, PF, and BC. Therefore, the complex, multidirectional interrelation between BC, PF, and PA and their correlates endorses that in addition to an active lifestyle, other risk factors that lead to poor BC, poor PF, or physical inactivity like hyper-caloric nutrition, malnutrition, or specific diseases, should be considered in further studies. Third, PA was assessed using self-reports, which are associated with several problems such as recall bias [99] and over- or underestimation of the activity level [100]. For this reason, device-based measurements such as accelerometry are often used and should be considered to replicate our findings regarding a relatively low influence of PA on BC. However, unlike accelerometry, self-reported PA can assess the setting in which PA takes place and involves a relatively low participation burden [101]. To obtain complete information about PA, both questionnaires and accelerometers should be used [99]. Lastly, it should be mentioned that the sample size was rather small despite high heterogeneity, especially concerning a relatively large range of age. Therefore, the results should be interpreted with caution.

On the other hand, our study has several strengths. Due to the representative sample drawn from the resident's bureau of a single model community, the results may be suitable to be generalized to individuals in middle and later adulthood living in Germany. BC was measured standardized using high-quality measuring equipment [49, 50]. A sophisticated test battery was utilized to measure motor performance and trained personnel were used for all measurements. With FMI, FFMI, and PhA as anthropometric parameters, a differentiated operationalization of BC was used in our study.

## Conclusion

Our results indicate that among individuals in middle and later adulthood, PF (cf. model 2) explains more variance of FMI, FFMI, and PhA compared to self-reported PA (cf. model 1). We found that strength is positively associated with FFMI and PhA among men and women and negatively associated with FMI only among women with strength and PhA showing the highest regression coefficients. This supports WHO's activity recommendations that adults should engage in resistance training to maintain an improvement in strength and to counteract age-related declines. Especially concerning Dynapenia, the age-related loss of strength plays a decisive role and should be considered when establishing PA recommendations. We also assume that PA might have a more noticeable effect on BC and PF if the intensity is emphasized in addition to the overall volume of PA. Accordingly, it is important to plan and structure the training sessions individually to achieve sustainable progression. However,

further investigations must be carried out to specify practical recommendations regarding volume and intensity in practice. Future research should examine the influences of PA and PF on BC using longitudinal analyses to draw causal conclusions and device-based measured PA should be used to replicate our results.

## Supporting information

**S1 Checklist. STROBE statement—checklist of items that should be included in reports of observational studies.**
(DOCX)

**S1 Fig. Scatter plots showing the relationship between age and PF.**
(TIF)

**S1 Table. Correlation matrix for age, BC, PA, and PF.**
(PDF)

## Author Contributions

**Formal analysis:** Raphael Schilling.

**Project administration:** Raphael Schilling.

**Supervision:** Raphael Schilling.

**Writing – original draft:** Raphael Schilling.

**Writing – review & editing:** Steffen C. E. Schmidt, Janis Fiedler, Alexander Woll.

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
