## [Decision Letter · Decision Letter 0]

2 Aug 2023

PONE-D-23-16245Associations between physical activity, physical fitness, and body composition in adults living in Germany: A cross-sectional studyPLOS ONE

Dear Dr. Schilling,

Thank you for submitting your manuscript to PLOS ONE. After careful consideration, we feel that it has merit but does not fully meet PLOS ONE’s publication criteria as it currently stands. Therefore, we invite you to submit a revised version of the manuscript that addresses the points raised during the review process. Please carefully respond to the reviewers' comments. PLOS ONE does not require novel findings for publication, and as such comments from reviewer one related to study novelty do not need to be directly addressed.  Although generally well-written, the conclusion paragraph is "rough" and needs some copy editing and clarifications. Please also be sure all relationships are directional (for example, "stronger" can mean a positive or negative relationship). 

We look forward to receiving your revised manuscript.

Kind regards,

Melissa M Markofski

Academic Editor

PLOS ONE

Reviewers' comments:

Reviewer's Responses to Questions

**Comments to the Author**

1. Is the manuscript technically sound, and do the data support the conclusions?

Reviewer #1: Partly

Reviewer #2: Partly

2. Has the statistical analysis been performed appropriately and rigorously? 

Reviewer #1: Yes

Reviewer #2: I Don't Know

3. Have the authors made all data underlying the findings in their manuscript fully available?

Reviewer #1: No

Reviewer #2: Yes

4. Is the manuscript presented in an intelligible fashion and written in standard English?

Reviewer #1: Yes

Reviewer #2: No

5. Review Comments to the Author

Reviewer #1: The paper “Associations between physical activity, physical fitness, and body composition in adults living in Germany: A cross-sectional study” aimed to describe the age-related development of body composition among adults and to examine the sex-specific associations between physical activity, physical fitness, and body composition. The study is well-structured and well-written. However, the paper does not present an original research question. Several other studies have already evaluated associations between body composition, level of physical activity, and physical fitness. Furthermore, the study included a low n for such a heterogeneous sample (men and women, 35 to 86 years).

Reviewer #2: In this research, the authors investigated the association between physical activity, physical fitness, and body composition. The findings of the study indicate that physical fitness exhibits a stronger association with body composition compared to physical activity. Moreover, the study suggests that muscle strength and cardiorespiratory fitness are better predictors of body composition. The authors discussed the key findings in the paper. However, significant improvements are required in the language to enhance clarity for the readers. Here are a few comments

Line 64- Please add reference after “routines”.

Line 69- You mentioned PA and lifestyle faster than ever. How Physical activity is faster in current scenario?

Line 73-76 “Although, except one ……describe BC”. This statement is very confusing. Please rewrite it clearly.

Line 77 – You mentioned that “there is a rationale for using FFMI and FMI….” Please describe the rationale in 2-3 sentences.

Line 104- Please do not use word elsewhere. You can use terms like previously or published in Woll et al…

Line 123- What is 5.25 in this calculation “5.25* 60”? How did you calculate this factor.

Line 162 – During z score transformation, did you use only men mean values to calculate both male and female or the female score was used for the women. In line 163 and 164, you mentioned only men. So, wanted to make sure.

Line 225- please add fig1

Line 284 – investigated constructs term is not clear. Please be specific.

Line 314-315 – I do not understand what you mean by “age and effect of strength and CRF supersede…….”

Line 318 – Remove the term straightforward

Line 319- is this entail or explain?

Line 322- Please do not use jargon like general increase. It is confusing.

Line 328- 329 – I do not understand what you are trying to say here.

Line 330 – What do you mean by evident link here? Please cite the paper.

Line 339- You mean large emphasis ‘given’ on….

Line 342-345- I do not understand your rationale. Please describe it clearly.

Line 351- 352 – “this could explain…” I am not sure what are you referring here. Please rewrite carefully.

Line 365 – Please clarify what do you mean by “certain intensity”.

Line 388- “two disease”- please add name of these disease.

Line 395- Please remove “ but not negative”. It is confusing.

6. PLOS authors have the option to publish the peer review history of their article (what does this mean?). If published, this will include your full peer review and any attached files.

Reviewer #1: No

Reviewer #2: No

---

## [Author Response · Author response to Decision Letter 0]

13 Sep 2023

After reviewing our manuscript, we made additional minor adjustments for better readability and consistency.

For a better understanding of the sample, an additional sentence was included in the results section.

(Page 8, Line 215)

“A total of 329 adults (183 women, 146 men) participated in our study.” 

Furthermore, we inserted consistent terms such as "in our study" to describe our work.

(Page 4, Line 89)

“In our study, we used a cross-sectional design to describe the age-related development of BC among adults and to examine the sex-specific associations between PA, PF, and BC.”

(Page 4, Line 94)

“Our cross-sectional analysis refers to the 2021 measurement point.”

(Page 4, Line 106)

“For our analyses, 88 subjects were excluded due to missing data in the variables of interest, and 13 subjects were excluded due to extreme values in the study variables.”

(Page 4, Line 108)

“Finally, a sample of 329 participants was examined in our study.”

(Page 7, Line 196)

“In our work, extreme values for men and women, respectively, were defined as data points that were more than 3 standard deviations away from the upper or lower limit of the interquartile range.”

(Page 13, Line 284)

“The goal of our study was to describe BC during adulthood using a cross-sectional sample from a German community and to investigate the associations between PA, PF, and BC under consideration of age.”

 (Page 13, Line 309)

“Studies also found that the rate of age-related loss of strength is higher than the decrease in muscle mass [32], which could explain why in our study, higher associations were found between age and strength compared to age and FFMI.”

(Page 15, Line 355)

“In our study, we found an association between sport activity and FMI only among women.”

 (Page 18, Line 449)

“There are limitations to our study.”

(Page 19, Line 464)

“On the other hand, our study has several strengths.”

Academic Editor:

Although generally well-written, the conclusion paragraph is "rough" and needs some copy editing and clarifications. Please also be sure all relationships are directional (for example, "stronger" can mean a positive or negative relationship).

Thank you for your comments on improving our paper. We have checked and reconsidered every case of effect sizes and (causal) relationships that we mention in the conclusion for objectivity. For a better understanding, we have also changed the structure of the Conclusion. We now use non-directional indicators of relationship when a set of parameters is related to another set of parameters (e.g. Physical Fitness on Body Composition) and the direction of those parameters is not homogeneous (e.g. positive relationship between Strength and FFMI, negative relationship with CRF). When single parameters are compared we only use directional indicators of the relationship. Below you can see the final version of the Conclusion:

(Page 19, Line 471-488)

“Our results indicate that among individuals in middle and later adulthood, PF (cf. model 2) explains more variance of FMI, FFMI, and PhA compared to self-reported PA (cf. model 1). We found that strength is positively associated with FFMI and PhA among men and women and negatively associated with FMI only among women with strength and PhA showing the highest regression coefficients. This supports WHO's activity recommendations that adults should engage in resistance training to maintain an improvement in strength and to counteract age-related declines. Especially concerning Dynapenia, the age-related loss of strength plays a decisive role and should be considered when establishing PA recommendations. We also assume that PA might have a more noticeable effect on BC and PF if the intensity is emphasized in addition to the overall volume of PA. Accordingly, it is important to plan and structure the training sessions individually to achieve sustainable progression. However, further investigations must be carried out to specify practical recommendations regarding volume and intensity in practice. Future research should examine the influences of PA and PF on BC using longitudinal analyses to draw causal conclusions and device-based measured PA should be used to replicate our results.”

Thank you for reminding us to review the style requirements. We have checked the style requirements and marked changes in the manuscript. Style changes were marked in green. The image files were uploaded to the digital diagnostic tool Preflight Analysis and Conversion Engine (PACE) and were adjusted. The file names have been adapted according to the PLOS ONE requirements.

The information about the ethical statement and procedure during testing can be found in the Method section. If there is still some information that is formally required from your Journal, please let us know and we will provide it as soon as possible.

(Page 4, Line 101-104)

 “Participation was voluntary and participants provided their written consent to participate in the study. The applied protocols were approved by a scientific advisory council, the Schettler Clinic, Bad Schönborn, Germany as well as the ethics committee of the Karlsruhe Institute of Technology. We strictly followed ethical guidelines from the German Psychological Society.”

We have checked our sources for completeness and checked each source to ensure that it has not been retracted. We could not find a retracted source. If you are referring specifically to individual sources, please feel free to let us know.

After checking all sources, we have exchanged 2 sources due to lack of informative value:

1.

Old Manuscript

World Health Organization. WHO guidelines on physical activity and sedentary behaviour. Geneva: World Health Organization; 2020. Available from: https://www.ncbi.nlm.nih.gov/books/NBK566045/.

New Manuscript

United Nations. World Population Ageing 2019-Highlights. New York: Department of Economic and Social Affairs; 2019. Available from: https://digitallibrary.un.org/record/3846855

2.

Old Manuscript

Trautner HM. Lehrbuch der Entwicklungspsychologie [Textbook of developmental psychology]. Göttingen: Hogrefe; 1978.

New Manuscript

Trautner HM. Zur Bedeutung der Begriffe Alter und Zeit in der Entwicklungspsychologie. In: Trautner HM, editor. Allgemeine Entwicklungspsychologie. 2nd ed. Stuttgart: Kohlhammer; 2003. pp. 31-35.

Furthermore, two additional sources were included in the manuscript:

3.

Ainsworth BE, Haskell WL, Whitt MC, Irwin ML, Swartz AM, Strath SJ, et al. Compendium of physical activities: an update of activity codes and MET intensities. Med Sci Sports Exerc. 2000;32(9): 498-504. doi: 10.1097/00005768-200009001-00009.

4.

Nuzzo JL. Narrative Review of Sex Differences in Muscle Strength, Endurance, Activation, Size, Fiber Type, and Strength Training Participation Rates, Preferences, Motivations, Injuries, and Neuromuscular Adaptations. Journal of strength and conditioning research. 2023;37(2): 494–536. doi: 10.1519/JSC.0000000000004329

Comments to the author:

Reviewer #1: The paper “Associations between physical activity, physical fitness, and body composition in adults living in Germany: A cross-sectional study” aimed to describe the age-related development of body composition among adults and to examine the sex-specific associations between physical activity, physical fitness, and body composition. The study is well-structured and well-written. 

We thank Reviewer 1 for pointing out the quality of our paper and for the helpful recommendations to further clarify some minor aspects. Please find a point-to-point response addressing all specific comments below.

1. The paper does not present an original research question. Several other studies have already evaluated associations between body composition, level of physical activity, and physical fitness.

We take note of the comment and agree with it to some extent. Nevertheless, it should be considered that with social changes and changes in everyday life, effects and relationships also change, and accordingly, studies on a specific topic should be carried out from time to time. One example is the WHO, which clearly states that permanent studies must be carried out to adapt and improve the activity recommendations.

2. The study included a low n for such a heterogeneous sample (men and women, 35 to 86 years).

We agree with the comment and added a statement to the limitations of this issue.

(Page 18-19, Line 461-463)

 “Lastly, it should be mentioned that the sample size was rather small despite high heterogeneity, especially concerning a relatively large range of age. Therefore, the results should be interpreted with caution.”

Reviewer #2: In this research, the authors investigated the association between physical activity, physical fitness, and body composition. The findings of the study indicate that physical fitness exhibits a stronger association with body composition compared to physical activity. Moreover, the study suggests that muscle strength and cardiorespiratory fitness are better predictors of body composition. The authors discussed the key findings in the paper. However, significant improvements are required in the language to enhance clarity for the readers. Here are a few comments

We thank reviewer 2 for his time and effort in assessing our paper and we address all comments below.

1. Line 64- Please add reference after “routines”.

Adjusted (Page 3, Line 64)

“While PA is defined by the movement of skeletal muscles which leads to energy expenditure, PF is expressed in daily life through various skills and routines [27].”

2. Line 69- You mentioned PA and lifestyle faster than ever. How Physical activity is faster in current scenario?

We removed the instance of “faster than ever” here because it did not add any information and is most likely wrong compared to different earlier periods of mankind. (Page 3, Line 69)

 “Nowadays, external influences like digital and social media as well as new technologies change our daily routines, PA, and lifestyle, making it important to analyze which behaviors are beneficial for successful aging.”

3. Line 73-76 “Although, except one ……describe BC”. This statement is very confusing. Please re-write it clearly.

Clarified as requested (Page 3, Line 77-79)

 “Most of these studies measure BC using bioelectrical impedance analysis [37, 38] or dual-energy X-ray absorptiometry [34, 35, 39, 40]. However, only a few studies use the fat mass index (FMI, as of fat mass (kg)/height2 (m)), and fat-free mass index (FFMI, as of fat-free mass (kg)/height2 (m)) to describe BC.”

4. Line 77 – You mentioned that “there is a rationale for using FFMI and FMI….” Please describe the rationale in 2-3 sentences.

Adjusted (Page 3-4, Line 82-86)

“Changes in BC are not necessarily clearly evident via the percentage of body fat, as both an increase in fat mass and a decrease in fat-free mass can result in the same percentage of body fat. For example, people with the same percentage of fat mass who differ in height may have a different nutritional state and/or fitness. In contrast, FMI and FFMI consider the amount of fat mass and fat-free mass in relation to body size [41].”

5. Line 104- Please do not use word elsewhere. You can use terms like previously or published in Woll et al…

Adjusted (Page 4, Line 111)

 “More details on the study procedures and additional measures have been published previously [43].”

6. Line 123- What is 5.25 in this calculation “5.25* 60”? How did you calculate this factor.

We clarified this in the method section. “5.25” refers to an average speed of walking for pleasure. (Page 5, Line 130-132)

“To obtain an average duration in minutes, the resulting average distances were multiplied by a factor of 5.25*60. The factor 5.25 results from the average speed of walking for pleasure according to Ainsworth and colleagues [45].”

7. Line 162 – During z score transformation, did you use only men mean values to calculate both male and female or the female score was used for the women. In line 163 and 164, you mentioned only men. So, wanted to make sure.

Exactly. We standardized according to men aged 35 at baseline. This leads to Z-values that are centered around an average man aged 35 at baseline. This method has some advantages when interpreting the null model and sex differences in hierarchical modeling but is a matter of preferences and the Z-values could have also been centered around the "average sex".

8. Line 225- please add fig1

The PLOS ONE guidelines state that figures should be uploaded as a separate file and that only a note should be included in the manuscript indicating where the figure needs to be inserted. Figure 1 was uploaded as a separate file called “Fig1.tif”.

9. Line 284 – investigated constructs term is not clear. Please be specific.

Adjusted (Page 13, Line 295)

 “In our study, the highest associations for both sexes were found for age and PhA.”

Line 314-315 – I do not understand what you mean by “age and effect of strength and CRF supersede…….”

To improve comprehension, we made changes in the last section of this point. (Page 14, Line 322-328)

 “Although the calendar age variable often shows strong correlations with parameters of lifestyle, fitness, and health, it should be noted that, from a theoretical point of view, time cannot be used to causally explain developmental trajectories [65]. There are always biological, psychological, or exterior processes and events that are responsible for age-related changes [66]. For example, a review concluded that a decrease in resting energy consumption caused by a decrease in organ mass and organ metabolism leads to increasing changes in BC with increasing age [21]. Thus, the effects of age should decrease when more explaining variables that influence the outcomes are added. This is also evident in our results. For both sexes, the regression coefficients of age decreased by adding fitness parameters to the model (Model 2.1).”

10. Line 318 – Remove the term straightforward

Adjusted (Page 14, Line 331)

 “In our cross-sectional analyses, there is no evidence that the total volume of sport activity or habitual activity is directly linked to BC”

11. Line 319- is this entail or explain?

This sentence clarifies that, according to our results, a higher self-reported physical activity level does not automatically lead to a better body composition. Therefore “entail” would be the right expression here. (Page 14, Line 332-333)

 “Higher self-reported activity levels do not directly entail a healthy BC in the form of low-fat mass and high levels of fat-free mass”

12. Line 322- Please do not use jargon like general increase. It is confusing.

Following the comment, we adjusted the entire manuscript.

Adjusted (Page 1, Line 13)

“To emphasize this, the World Health Organization established activity recommendations for all age groups.”

Adjusted (Page 5, Line 117)

“In addition to the question "Do you engage in sports or gymnastic exercises?", the types of sports were asked, as well as the minutes per week and the number of weeks per year in which the activity is pursued.”

Adjusted (Page 14, Line 335)

 “It revealed that an increase in the activity level does not lead to health benefits, such as a reduction in BMI”

Adjusted (Page 18, line 443)

“However, a decrease in activity level also has a negative effect on coordination ability [97].”

13. Line 328- 329 – I do not understand what you are trying to say here.

We added a sentence for clarification. (Page 15, Line 346-348)

“It is assumed that PF acts as a mediator between PA and different health parameters such as overweight and subjective as well as diagnosed orthopedic and psychosomatic impairments [29, 68]. That means PA that does not lead to significant changes in PF may not result in health benefits and that amounts and intensities of PA that lead to a decrease in PF (overtraining) can also lead to a decline in health.”

14. Line 330 – What do you mean by evident link here? Please cite the paper.

The clarification of the sentence above made this sentence redundant and we removed it. We referred to each study that found a significant correlation between PF and BC.

15. Line 339- You mean large emphasis ‘given’ on….

Adjusted (Page 15, Line 357)

 “A larger emphasis given on health and body weight [76, 77] instead of physical improvement, muscle mass, and strength during the choice and execution of sports as well as the linked lifestyle and eating patterns among women [78] could be one reason for this outcome.”

16. Line 342-345- I do not understand your rationale. Please describe it clearly.

Clarified (Page 15, Line 359-362)

 “Reasons for a missing association between PA and FMI among men may be the underlying motive of exercising. A potential loss of fat mass may be prevented by an increase in appetite and food intake when exercising primarily for improving strength and muscle mass which is usually preferred by men [79].”

17. Line 351- 352 – “this could explain…” I am not sure what are you referring here. Please rewrite carefully.

For a better understanding, we have rewritten the paragraph. (Page 16, Line 367-393)

 “A recent review found that in females, the total volume of PA plays a more important role in health benefits than intensity [80]. Especially concerning metabolic syndrome, which is also defined by an immoderate amount of body fat [81], two studies identified a positive association with the volume of PA [82, 83] whereas another study found positive effects only between moderate to vigorous PA and metabolic syndrome [84]. According to this research, women should choose a higher volume of PA to achieve health effects such as a decrease in FM. These circumstances might be another reason why we found a significant negative relationship between the volume of sport activity and FMI in women. In addition to the combination of load intensity and volume of PA, the duration of every single bout of PA also seems to have a decisive influence. In a recent review, it was concluded that despite high intensities and an associated increase in physical performance, short exercise bouts between 60 and 240 seconds did not lead to a reduction in body fat [85].

In conclusion, our results regarding PA and BC indicate that in women, a higher volume of sport activity is related to a lower FMI value. Discussing the results in the context of the current state of research, we assume that high amounts of PA are more effective in women, especially in reducing FM. Aerobic activities are effective for the reduction of body fat but have a lower impact on fat-free mass or PhA. Strength training could be decisive here. Improving BC through PA, therefore, requires a combination of endurance and weight training, as well as a periodization of training. This supports the activity recommendations of the WHO.”

18. Line 365 – Please clarify what do you mean by “certain intensity”.

For a better understanding, we removed this sentence from the paragraph above.

19. Line 388- “two disease”- please add name of these disease.

Adjusted (Page 17, Line 411-412)

“A recent review has revealed that resistance training provides a significant impact on the progression and outcome of sarcopenia and frailty”

20. Line 395- Please remove “but not negative”. It is confusing.

Adjusted (Page 17, Line 418)

 “In a study among elderly persons in Spain, a cross-sectional design demonstrated a positive correlation coefficient between VO2max and body weight in women [93].”

---

## [Editor Report · Decision Letter 1]

16 Oct 2023

Associations between physical activity, physical fitness, and body composition in adults living in Germany: A cross-sectional study

PONE-D-23-16245R1

Dear Dr. Schilling,

We’re pleased to inform you that your manuscript has been judged scientifically suitable for publication and will be formally accepted for publication once it meets all outstanding technical requirements.

Kind regards,

Melissa M Markofski

Academic Editor

PLOS ONE
---

## [Editor Report · Acceptance letter]

19 Oct 2023

PONE-D-23-16245R1 

Associations between physical activity, physical fitness, and body composition in adults living in Germany: A cross-sectional study 

Dear Dr. Schilling:

I'm pleased to inform you that your manuscript has been deemed suitable for publication in PLOS ONE. Congratulations! Your manuscript is now with our production department. 

Kind regards, 

on behalf of

Dr. Melissa M Markofski 

Academic Editor

PLOS ONE